# Zwitterionic molecularly imprinted polymers for selective capillary microextraction of N1,N12-Diacetylspermine (DiAcSpm) from breast cancer

**Keming Ying**[1]*, **Han Xue**[1], **Shenheng Du**[1], **Xin Wang**[2], **Xiaoyun Lei**[2]

1 Department of Surgical Oncology, Hanzhong Central Hospital, Hanzhong, Shaanxi Province, China,
2 Shaanxi Key Laboratory of Catalysis, School of Chemical and Environmental Science, Shaanxi University of Technology, Hanzhong, China

* kemingying0912@126.com

## Abstract

N1,N12-diacetylspermine (DiAcSpm), a promising biomarker for cancer diagnosis, presents significant quantification challenges due to the structural homology within the polyamine family. To address this issue, we engineered a molecularly imprinted monolithic (MIM) column functionalized with biomimetic phosphorylcholine (PC) as functional monomer for the selective recognition of DiAcSpm in human urine. The zwitterionic polymer was synthesized via thermally initiated polymerization, with its morphology and pore architecture characterized through scanning electron microscopy (SEM) and brunauer-emmett-teller (BET) analysis. After optimizing capillary microextraction (CME) parameters, the MIM demonstrated a broad linear response (10–500 μM), a low detection limit (3.3 μM, S/N = 3), and high recoveries (76.8–91.2%) when coupled with HPLC-UV analysis. The biomimetic PC-based recognition significantly improved selectivity against key structural analogs, such as spermine, in complex biological matrices. This study underscores the potential of zwitterionic-based MIMs as a robust and efficient platform for the sensitive and selective monitoring of acetylated polyamines in clinical settings.

## 1. Introduction

Breast cancer, a prevalent hormone-related malignancy, is primarily influenced by two significant risk factors: female gender and age [1,2]. By 2040, the incidence of breast cancer is projected to increase by over 40%, reaching approximately 3 million cases annually due to population growth and aging [3]. In China, breast cancer has become one of the most common cancers among women, with its incidence rate steadily increasing over the past five years. Current therapeutic strategies typically involve a combination of surgery, chemotherapy, radiotherapy, and hormone therapy [4–6]. Despite substantial advancements in understanding the molecular basis of breast cancer over the last two decades, there remains a critical unmet need for the

**Data availability statement:** All relevant data are within the paper and its Supporting Information files.

**Funding:** This work was funded by the Hanzhong Science and Technology Research Project (SKJJKJGG05); Doctoral Studio Program of Hanzhong Central Hospital (YGS24-06).

**Competing interests:** The authors have declared that no competing interests exist.

development of reliable biomarkers for early detection and treatment monitoring [7,8]. As a result, identifying effective methods for early screening and treatment evaluation through straightforward experimental procedures has become increasingly important.

Polyamines, particularly diacetylated spermine (DiAcSpm), have emerged as pivotal players in cancer biology due to their roles in cellular proliferation and oncogenic pathways, such as the phosphatase and tensin homolog-phosphatidylinositol 3-kinase-mechanistic target of rapamycin (PTEN-PI3K-Mtor), wingless/Int-1 (WNT) and rat sarcoma virus (RAS) signaling pathways [9–11]. Urinary N1,N12-diacetylspermine (DiAcSpm) is significantly elevated in breast cancer patients and possesses considerable potential as a non-invasive diagnostic biomarker, with an estimated cut-off of 150–850 nM and a sensitivity of 46.4% that surpasses conventional serum biomarkers, as measured by competitive ELISA (Enzyme-Linked Immunosorbent Assay) [12]. Unlike traditional biomarkers that require invasive sampling, the quantification of spermine in urine offers a simple yet powerful strategy for early detection and monitoring of therapy [13,14]. However, there are still challenges in spermine detection, such as the high polarity of spermine leads to poor retention on reversed-phase columns on high-performance liquid chromatography (HPLC) and liquid chromatography-tandem mass spectrometry (LC-MS/MS) [15–17]. Derivatization or solid-phase extraction (SPE) pre-treatment steps increase time and resource demands [10,18]. Moreover, complex biological samples, such as urine, introduce co-eluting substances that compromise accuracy. While molecularly imprinted polymers (MIPs) could enhance specificity by selectively binding spermine, conventional MIPs struggle with matrix effects (MEs) and stability in aqueous environments [19–22]. MEs arise from the presence of co-eluting substances in biological samples, which can interfere with the ionization process, causing signal bias and ultimately impacting result accuracy.

To mitigate the impact of matrix effects on analytical results, the utilization of biocompatible materials as functional monomers in molecular imprinting has emerged as a significant strategy [23,24]. Zwitterionic materials, as a subset of biocompatible materials, not only enhance the selectivity and affinity of molecularly imprinted polymers but also minimize interference from matrix components, thereby improving the detection accuracy of target molecules [25]. 2-Methacryloyloxyethyl phosphorylcholine (MPC) is a methacrylate mimetic monomer featuring a phosphorylcholine (PC) moiety on its side chain [26]. It possesses an amphoteric ionic structure consisting of a phosphate anion and a trimethylammonium cation, which facilitates strong electrostatic interactions [27]. These zwitterionic materials exhibit unique properties that render them highly effective as ultralow fouling materials in various biomedical and engineering applications [28]. The head group of MPC polymers demonstrates significant hydration due to these electrostatic interactions, resulting in pronounced hydrophilicity. Consequently, they are frequently employed in chromatography for the preparation of hydrophilic interaction liquid chromatography (HILIC) stationary phases [29,30]. Furthermore, their exceptional resistance to non-specific protein adsorption positions them as promising candidates for mitigating complex matrix interferences in biological environments [31]. These distinctive properties establish

MPC as a hydrophilic amphiphilic functional monomer suitable for the development of molecularly imprinted polymers (MIPs) for aqueous recognition. Although a limited number of studies have investigated the application of amphiphilic ions as functional monomers or co-monomers in the development of aqueous-recognition MIPs [32,33], the existing research underscores their potential significance.

In this work, we efficiently synthesis of a zwitterionic hybrid monolithic column via a one-pot copolymerization strategy within a capillary, utilizing 2-methacryloyloxyethyl phosphorylcholine (MPC). This monolithic structure was designed as a molecularly imprinted polymer (MIM) for the selective capture of the cancer biomarker N1,N12-diacetylspermine (DiAc-Spm) in complex biofluids. After optimizing the synthesis protocol, the material's morphology, elemental composition, and porosity were characterized using scanning electron microscopy (SEM), EDS elemental mapping and nitrogen adsorption-desorption analyses. Prior to application, the operational parameters for the MIM-based microextraction coupled online with HPLC-UV were optimized. The established method was then successfully deployed to quantify DiAcSpm in urine samples from individuals with breast cancer and healthy volunteers.

## 2. Materials and methods

### 2.1. Reagents

N1, N12-diacetylspermine (DiAcSpm), spermidine, spermine and norspermine were all purchased from Aladdin Reagent (Shanghai, China). 2-Methacryloyloxyethyl phosphorylcholine (MPC) were purchased from TCI Development Co., Ltd. (Shanghai, China). Ethylene glycol dimethacrylate (EDMA) and 3- (trimethoxysilyl) propyl methacrylate (γ-MAPS) were purchased from J&K Scientific Ltd. (Beijing, China). Azobisisobutyronitrile (AIBN) was purchased from Shanghai Reagent No.4 Factory (Shanghai, China). Methanol (MeOH) and tetrahydrofuran (THF) were purchased from Shanghai Reagent Chemical Co., Ltd. (Shanghai, China). HPLC-grade acetonitrile (ACN) and trifluoroacetic acid were obtained from Sigma-Aldrich (St. Louis, MO, USA). Pure water used in the laboratory was obtained through the Milli-Q water purification system (18.2 M•cm, Millipore, Bedford, MA, USA). The fused silica capillaries (530 μm i.d. × 690 μm o.d.) were purchased from Hebei Yongnian Optical Fiber Factory (Hebei, China).

### 2.2. Instruments

The chromatographic analysis was performed using a Shimadzu SPDM20A HPLC system (Kyoto, Japan), which included two LC-20 CE pumps, a SIL-20 A autosampler, an SPD-M20A diode array detector, and a CTO-20 AC column oven. A Caprisil C18-X column (250 mm × 4.6 mm i.d., 5.0 μm) was utilized at 30 °C to separate the analytes (Morhchem Technologies, Beijing, China). UV detection was set to a wavelength of 210 nm. Mobile phase A, used for HPLC gradient elution, consisted of water with 0.1% trifluoroacetic acid (v/v), while mobile phase B was acetonitrile. The analytes were separated using the following gradient: initially, a 10.0 min step was performed with a 5% mobile phase B gradient up to 30% B, which then remained constant for 15.0 min. The surface morphologies and the energy spectrum analysis (EDS) of the MIM and NIM columns were characterized by SEM (JEOL JSM-IT300, Japan). Pore size distribution of MIM and NIM columns were analyzed by physical adsorption and desorption instrument (ASAP 2460, Micromeritics Instrument Corp., USA).

### 2.3. Preparation of molecularly imprinted monolithic (MIM) column

The preparation of the MIM columns involved two primary steps. First, the inner wall of the capillary column was pre-treated to alkenylate the exposed surface, following a method previously described in our earlier work [34]. As shown in Fig 1, a homogenized pre-polymerization mixture was prepared, containing 30.0 mg of functional monomer (MPC) and template (DiAcSpm), with a molar ratio of template to monomer was 1:9. A binary porogenic agent composed of 76.0 μL of methanol (MeOH) and 270.0 μL of tetrahydrofuran (THF) was selected to dissolve the reagents. The mixture was then subjected to ultrasonication after vortexing and shaking for 30.0 min to form a template-monomer

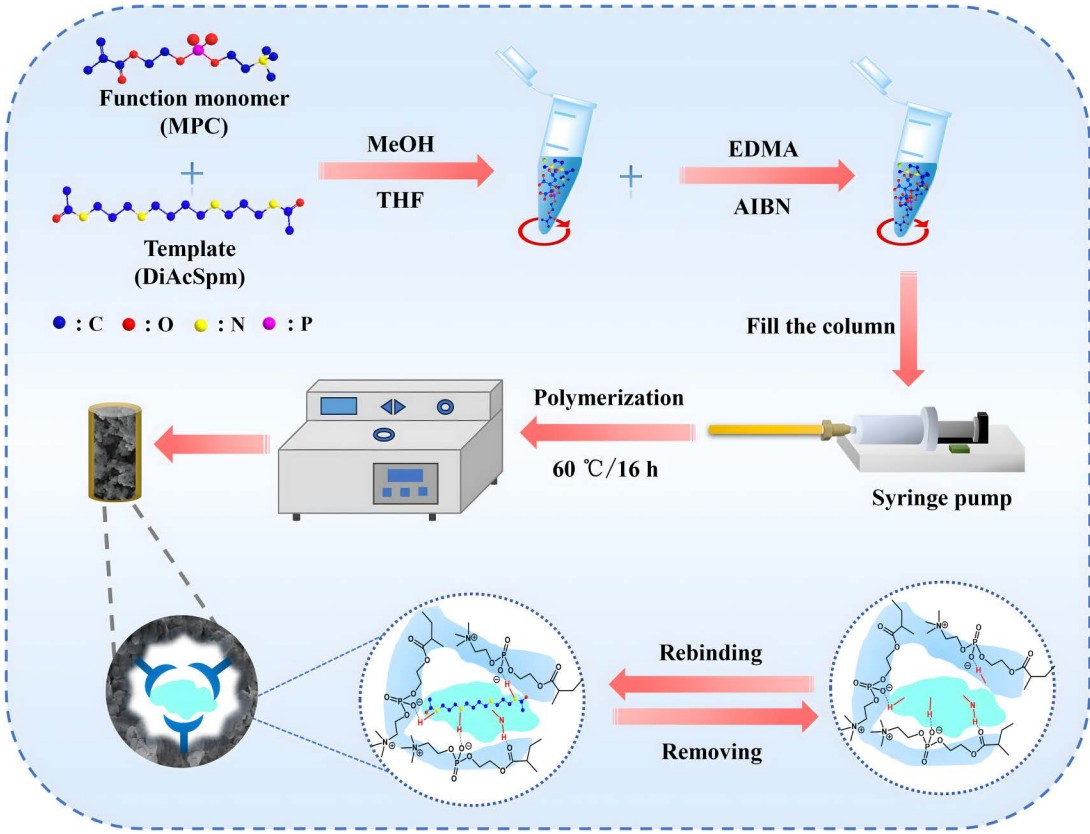

**Fig 1. The preparation of zwitterionic-based molecularly imprinted monolithic (MIM) column.**

complex prepolymerization solution. In the next step, the cross-linking agent EDMA (30.0 mg) and the initiator AIBN (1.0 mg) were added and the mixture was thoroughly mixed and ultrasonicated for 20.0 min to eliminate bubbles. The solution was purged with nitrogen for 5 min to remove oxygen before polymerization. Then, the prepared solution was slowly injected into the pretreated capillary using a syringe pump, sealed at both ends with a plunger, and then reacted in a water bath at 60 °C for 16 h. After polymerization, the MIM column was rinsed for over 30.0 min using washing solution of MeOH/H$_2$O (1:9 v/v) mixture to remove residual reagents and templates. Non-molecularly imprinted monolithic (NIM) columns were prepared in the same manner, except that the template molecules were omitted.

## 2.4. Capillary microextraction (CME) analysis

The capillary microextraction (CME) process was conducted using a high-pressure constant-flow pump (Series II, LabAlliance, State College, USA) connected to a 3.0 mL PEEK chromatography quantitative ring. Prior to each CME run, the MIM/NIM column was activated by passing 1.0 mL of a MeOH/H$_2$O (1:9, v/v) followed by methanol, at a flow rate of 50.0 µL min$^{-1}$. Subsequently, the 3.0 mL quantification ring, containing varying concentrations of DiAcSpm loading solvent (80% (v/v) ACN/H$_2$O), was passed through the monolithic column at the same flow rate of 60.0 µL min$^{-1}$. A wash solution of methanol/water (1:1, v/v) was then employed to rinse the monolithic column and eliminate interferences. The analytes were retained on the columns and eluted with 50 µL of eluent (10% ACN/0.2% (v/v) formic acid in water) at a flow rate of 70.0 µL min$^{-1}$. Finally, the collected eluent was filtered through a 0.22 µm filter membrane prior to HPLC-UV analysis.

## 2.5. Enrichment performance of MIMs and NIMs

A 20 μM of N1, N12-diacetylspermine (DiAcSpm) standard solution was flowed through the microextraction columns at a constant flow rate of 60.0 μL min$^{-1}$. The elution obtained was collected and then derivatized for HPLC-UV detection. The enrichment abilities ($Q_c$) of the MIM and MIM columns were determined using the equation outlined in our previous publication [35].

$$Q_c = ctv/m$$

Where $Qc$ (μmol/g) is adsorption capacity, $c$ is the concentration of DiAcSpm in the eluent (μM), $v$ is the volume of the eluent (mL) and $m$ is the mass of the monolithic column (g).

## 2.6. Pretreatment of urine samples

Urine samples were randomly collected from breast cancer patients and healthy controls from Hanzhong Central Hospital. Prior to analysis, each 5 mL urine sample was centrifuged at 3000 × g for 10 min at 4 °C to remove precipitates and cellular debris. Subsequently, 3 mL of the supernatant was aliquoted and stored at −80 °C until further analysis. A stock standard solution of DiAcSpm (approximately 1.0 mg/mL) was prepared by dissolving an accurate mass of the solid standard in 1.0 mL of ultrapure water. Then the solution was stored in refrigerator at −20 °C in the dark. Working standard solutions at desired concentrations were obtained by serial dilution of the stock solution with ultrapure water or an appropriate buffer. For the preparation of spiked samples, a known volume of the working standard solution was added directly to the processed urine supernatant. All solutions were filtered using a 0.22 μm filter membrane for HPLC-UV analysis. This study involving human participants was approved by the Institutional Review Board (IRB) of the Central Hospital of Hanzhong. Verbal informed consent was obtained from all participants recruited between August 2024 and February 2025, with documentation via a standardized form. This form was read aloud, completed by the researcher upon the participant's verbal agreement, and countersigned by an independent staff member to validate the voluntary nature of consent. All signed forms are maintained as confidential records.

## 3. Result and discussion

### 3.1. Fabrication of zwitterionic MIM column

In this study, a zwitterionic molecularly imprinted material (MIM) column was developed for the specific recognition of DiAcSpm through in situ polymerization within a capillary tube. MPC was selected as the zwitterionic function monomer owing to its exceptional resistance to nonspecific protein adsorption and excellent biomimetic properties, which are crucial for achieving selective extraction from complex biological matrices such as urine. The molar ratio of template to monomer is critical in the formation of molecularly imprinted polymers. An optimal monomer concentration facilitates the formation of hydrogen bonds between the template and the polymer, resulting in the creation of suitable binding sites. Conversely, an excess of monomer could lead to increased non-specific affinity. In this work, the template-monomer molar ratio was optimized, as detailed in Fig 2A. The enrichment efficiency exhibited a gradual increase as the molar ratio of template to monomer decreased from 1:4.5 to 1:9. However, the efficiency decreased when the molar ratio was further decreased from 1:15–1:20. This decline is likely attributable to the requirement for sufficiently high concentrations of functional monomers to ensure a high binding capacity for target molecules, as excessive monomer concentrations tend to enhance their own aggregation, thereby reducing adsorption capacity. Consequently, a template-monomer molar ratio of 1:9 was selected as optimal for subsequent studies.

The selection of an appropriate washing solvent is critical for minimizing nonspecific interactions and enhancing the specificity of molecularly imprinted monolithic (MIM) column by effectively removing residual template molecules and

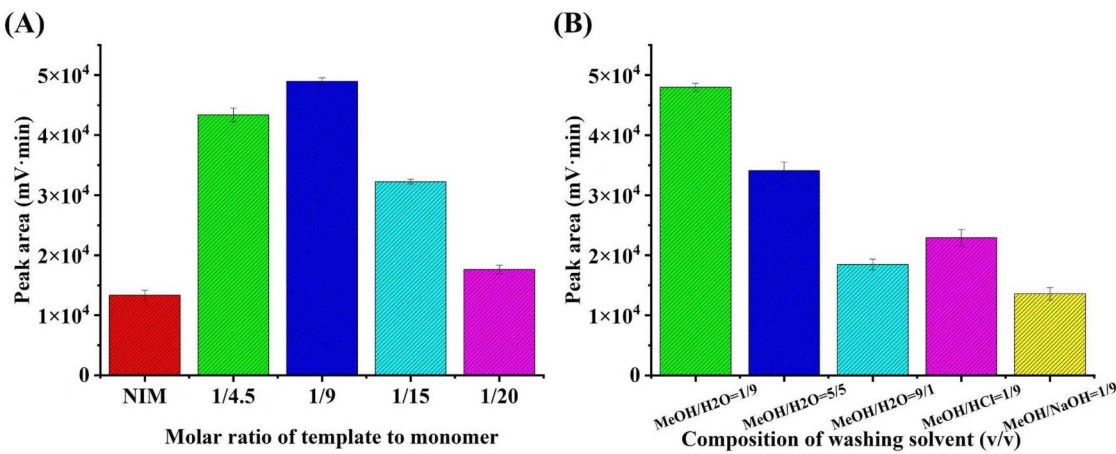

**Fig 2. Effect on (A) the molar ratio of the template to the monomer and (B) the composition of washing solvent (v/v).**

interfering substances [36]. As illustrated in Fig 2B, increasing the MeOH to $H_2O$ ratio in the washing solvent from 1:9–9:1 led to a gradual reduction in enrichment performance. This trend suggests that excessive methanol content weakens the hydrophobic and hydrogen-bonding interactions that are critical for analyte retention. Furthermore, the combination of methanol with hydrochloric acid (HCl) or sodium hydroxide (NaOH) resulted in the inefficient removal of the template molecule from the biomimetic extraction material. Acidic or alkaline additives likely induced protonation or deprotonation of functional groups on both the polymer and the template, thereby altering electrostatic interactions and compromising the stability of the imprinted cavities. In conclusion, the MeOH/$H_2O$ (1:9, v/v) optimally balances the removal of nonspecifically adsorbed interferents while preserving the integrity of the imprinted sites, thereby ensuring robust analyte enrichment.

### 3.2. Characterization of the MIM column

In this experiment, scanning electron microscopy (SEM) was employed to analyze the surface morphology of NIM and MIM columns, as illustrated in Fig 3. The results demonstrate that both columns exhibit a continuous porous structure with an average diameter of approximately 5.0 μm, which facilitates efficient mass transfer. Fig 3A and 3B present the high-magnification polymer morphology of the NIM column, revealing a relatively smooth particle surface. In contrast, Fig 3C and 3D depict the morphology of the MIM column prior to template removal, showing a comparatively rough surface. This roughness may be attributed to the strong adhesion of the templates to the material surface. Following the washing and removal of the templates, cavities that can specifically bind to the target analytes will be exposed, thereby enhancing the capture efficiency of the analytes.

The results of $N_2$ adsorption-desorption isotherms and pore size distribution curves (insert) of NIM and MIM polymers are presented in Fig 4A and 4B. Both samples exhibit type IV curves with hysteresis loops and narrow pore size distributions, indicating the presence of regular channel structures. The pore structural parameters further elucidate their differences: the MIM possesses a specific surface area of 28.11 $m^2$/g, an average pore diameter of 12.34 nm, and a total pore volume of 0.089 $cm^3$/g. In contrast, the NIM shows lower values, with a specific surface area of 16.22 $m^2$/g, an average pore diameter of 8.76 nm, and a total pore volume of 0.052 $cm^3$/g. The notably larger pore size and greater pore volume of the MIM are attributed to the successful removal of template molecules, which creates imprinted cavities within the polymer matrix. These structural characteristics suggest that the MIM could serve as an excellent adsorption and recognition carrier. Moreover, the energy dispersive spectroscopy (EDS) of hybrid monolith (S1 Fig) shows the presence of elements N and P, which indicates the incorporation of MPC monomer in the MIM column matrix.

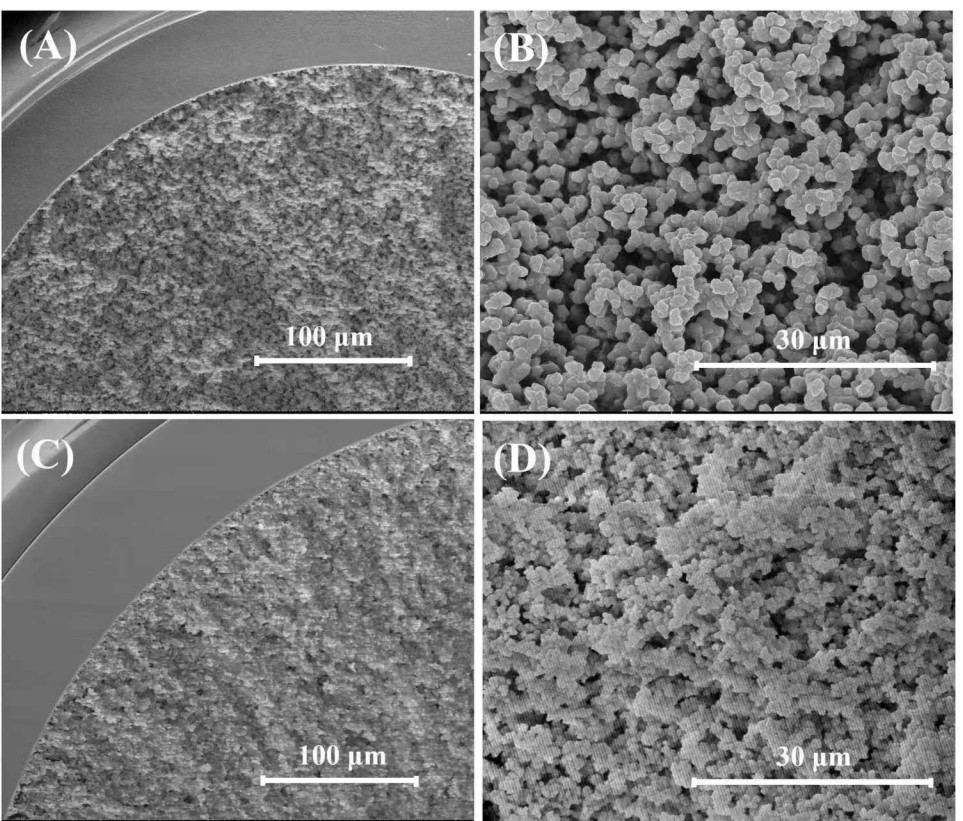

**Fig 3. SEM micrographs of (A and B) NIM and (C and D) MIM column before template removal.** Magnification: A and C: 1,000×; B and D: 5,000×.

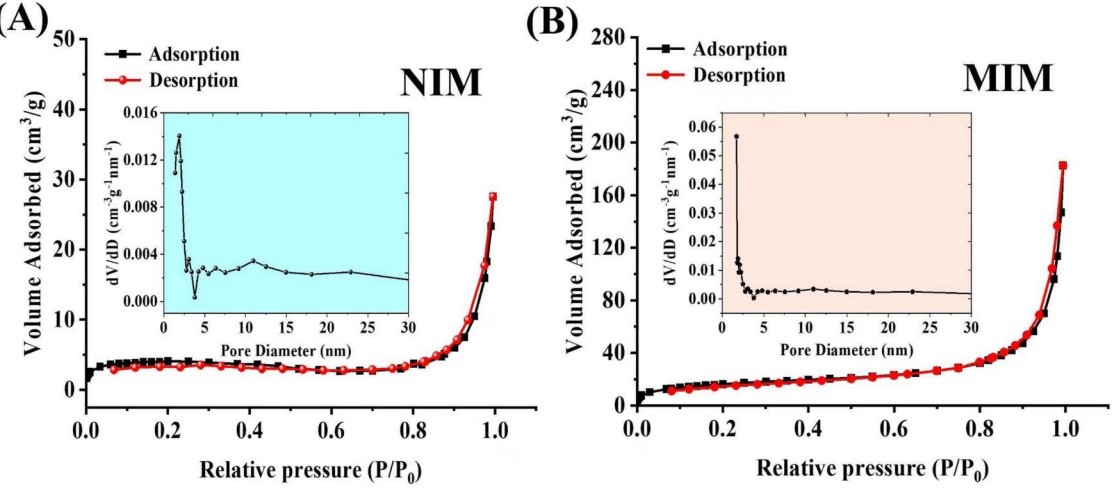

**Fig 4. Specific surface area and pore size distribution curve (insert) of the (A) NIM and (B) MIM column.**

### 3.3. Optimization of the extraction conditions

**3.3.1. Optimization of the loading solvent composition.** In this experiment, acetonitrile (ACN) was utilized as the loading solvent in the organic phase. The ratio of ACN was optimized, with the results presented in Fig 5A. The extraction performance of the MIM column for the target analyte gradually improved as the ACN content increased from 50% to 80%. This enhancement can be attributed to the reduction in water content accompanying the increased ACN concentration. Consequently, the polar solvent content was minimized, thereby reducing potential damage to the non-covalent specific interactions between the template and the MIM column [37]. However, when the loading solvent was composed of pure ACN, the decreased solubility of DiAcSpm in the solution resulted in a lower affinity. Ultimately, an optimal enrichment ratio of 80% (v/v) ACN/$H_2O$ was established.

**3.3.2. Optimization of the eluent composition.** The composition of the eluent is critical for the enrichment efficiency of molecularly imprinted materials, as it governs the hydrophobic interactions and solvation effects between the target molecule and the imprinted cavities. In this study, a series of eluents with varying proportions of acetonitrile (ACN) ranging from 0% to 50% (v/v) were evaluated, while maintaining a constant concentration of 0.2% (v/v) formic acid in water. As illustrated in Fig 5B, the extraction capacity of DiAcSpm initially increased with the ACN content, reaching a maximum at 10% ACN. This trend can be attributed to the balanced solvation and hydrophobic interactions, where a moderate organic phase weakened the polar interactions between DiAcSpm and the aqueous matrix while maintaining sufficient affinity for the hydrophobic cavities of the imprinted polymer. At higher ACN concentrations (e.g., 20–50%), the excessive

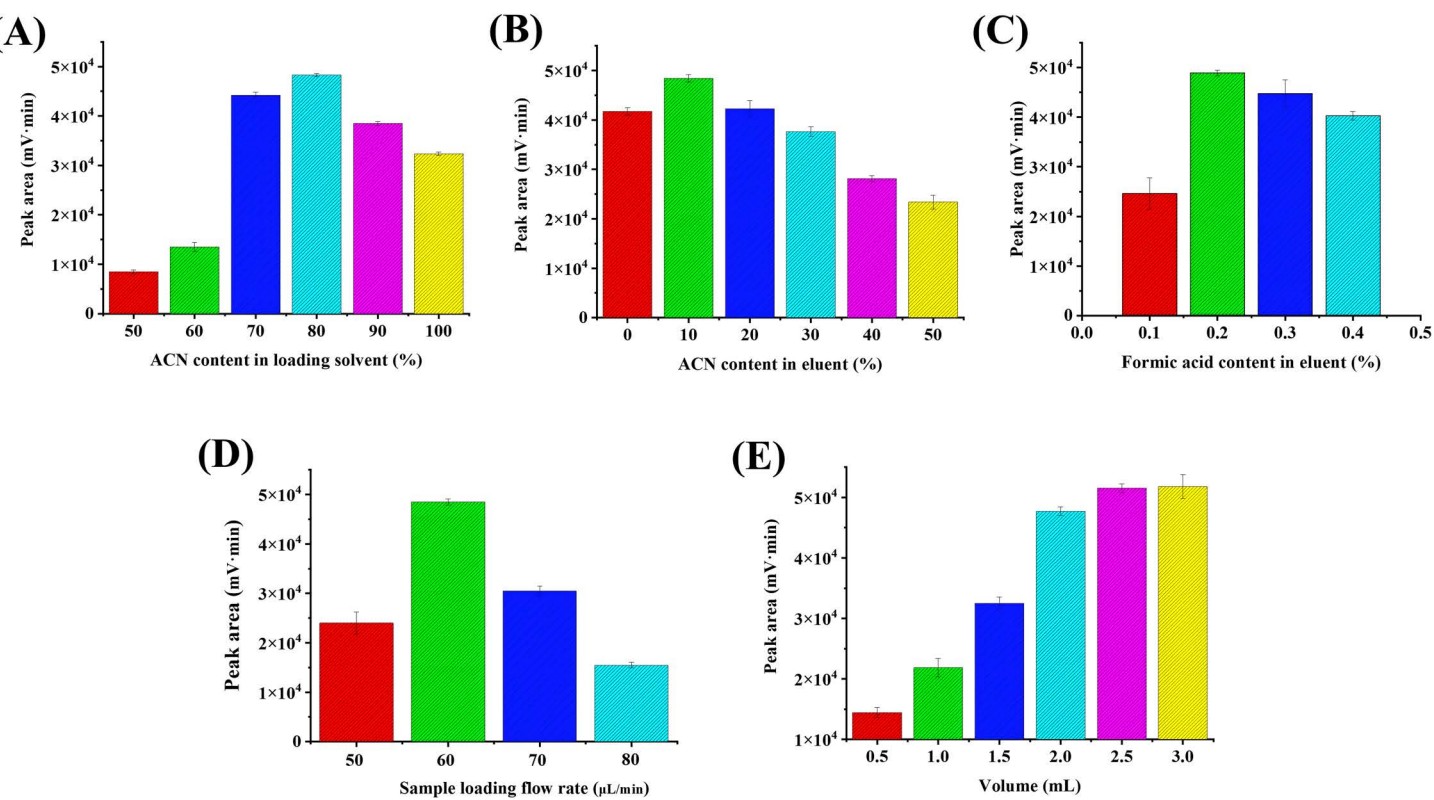

**Fig 5. Effects of MIM-CME conditions on the extraction performance of DiAcSpm. (A)** ACN content in loading solution (%); **(B)** ACN content in eluent (%); **(C)** formic acid content in eluent (%); **(D)** flow rate of loading solvent (μL/min); **(E)** volume of sample loading (mL). All experiments were performed in triplicate (n = 3).

hydrophobicity of the eluent likely disrupted the specific interactions between DiAcSpm and the imprinted sites, favoring non-specific binding or incomplete desorption. Additionally, the reduced polarity of the eluent may have compromised the stability of hydrogen bonds formed during the imprinting process. Consequently, 10% ACN/0.2% (v/v) formic acid in water was identified as the optimal ratio, achieving a compromise between effective elution strength and the preservation of selective recognition mechanisms.

### 3.3.3. Optimization of pH on eluent.

Furthermore, the pH of the eluent, governed by the formic acid content, significantly influences the desorption process. Although the zwitterionic monomer MPC maintains a net negative charge over a wide pH range, facilitating cation-exchange interactions, the strength of these interactions is modulated by the ionic environment. As shown in Fig 5C, the peak area of DiAcSpm initially increased as the formic acid content rose from 0.1% to 0.2%, reaching a maximum. This could be explained by the sufficient protonation of DiAcSpm, which strengthens its ionic interaction with the stationary phase during the loading step and allows for efficient release in a mild acidic eluent. However, beyond 0.2%, a higher formic acid content (0.3%–0.5%) resulted in a gradual decrease in the peak area. This decline is likely due to an excessively acidic environment causing overly strong protonation or altering the solvation equilibrium, thereby hampering complete desorption. Based on these observations, an eluent consisting of 0.2% formic acid in eluent solution was identified as the optimal condition.

### 3.3.4. Optimization of the sample loading flow rate.

The effect of sample loading flow rate was examined in this work (Fig 5D). It was observed that the adsorption efficiency of DiAcSpm decreased when the flow rate exceeded from 60 µL/min to 80 µL/min. This decrease was attributed to the shorter interaction time between DA and the recognition site of the MIM column at higher flow rates. On the other hand, the reduction in adsorption efficiency at lower flow rates was attributed to the desorption of the adsorbent in the MIM columns. Consequently, a flow rate of 60 µL/min was determined to be the optimal loading flow rate.

### 3.3.5. Optimization of the sample loading volume.

The loading volume of the enrichment solution is a critical parameter that governs the adsorption capacity and efficiency of molecularly imprinted materials (MIMs) for target analytes. Theoretically, an increase in the loading volume enhances the total amount of analytes available for binding, thereby improving enrichment performance proportionally. However, experimental results in Fig 5E have revealed that increasing the sample volume from 2.5 mL to 3.0 mL does not yield a corresponding twofold improvement in enrichment efficiency, indicating saturation of the accessible binding sites on the MIMs. This nonlinear behavior can be attributed to the finite density of imprinted cavities and the dynamic equilibrium between analyte adsorption and desorption. At lower volumes, analytes can fully occupy available binding sites due to sufficient interaction time and unrestricted mass transfer. However, when the volume exceeds this critical point (e.g., 3.0 mL), the limited number of high-affinity imprinted cavities becomes saturated, resulting in excess analytes remaining unbound due to insufficient specific adsorption capacity. Therefore, the final optimal sample loading volume was established as 2.5 mL.

## 3.4. Performance characterization of the NIM/MIM column

### 3.4.1. Extraction capacity of NIM and MIM columns.

The loading capacity of the MIM and NIM columns were evaluated by the adsorption capacity ($Q_c$) and the details of the calculation were partially outlined in Section 2.5. A specific concentration 20 µM of DiAcSpm was selected for analysis of adsorption capacity using the MIM-CME-HPLC-UV system. As illustrated in Fig 6A, the adsorption amounts for both the MIM and NIM columns increased with time. The MIM column exhibited an adsorption capacity of 67.45 ± 2.84 µmol/g, significantly higher than that of the NIM column, which recorded 38.54 ± 1.92 µmol/g across the experimental concentration range. This difference was statistically confirmed by an unpaired t-test ($p < 0.001$, $n = 5$), indicating the successful creation of specific binding sites in the MIM.

### 3.4.2. Selectivity of MIM column.

In this study, spermine, spermidine, and norspermine were employed as structural analogs for the MIM-CME selectivity experiments. Their molecular structures are illustrated in S2 Fig, and a comparative analysis of their key structural features relative to the template and the target analyte (DiAcSpm) is provided in S1 Table.

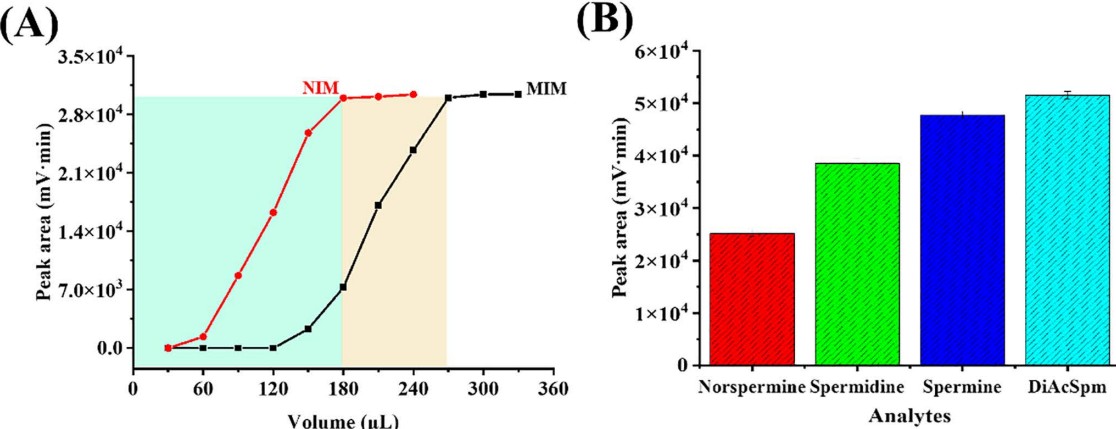

**Fig 6. Evaluation of the extraction performance of MIM and NIM columns for DiAcSpm.** (A) The loading capacity of the MIM and NIM columns for the specific extraction of DiAcSpm; (B) selectivity of MIM and NIM columns for DiAcSpm and its structural analogues. The concentration of all analytes was 25 µM.

As illustrated in Fig 6B, the MIM column demonstrated significantly higher selectivity for DiAcSpm compared to the other compounds. This enhanced specificity can be attributed to the unique diacetylation modification of DiAcSpm, which facilitates stronger hydrogen bonding and hydrophobic interactions with the functional groups on the MPC (e.g., phosphate group) present in the MIM material. In contrast, unmodified polyamines such as spermine and spermidine lack acetyl groups, leading to reduced binding affinity due to steric hindrance and polarity mismatches. Moreover, the imprinted cavities within the MIM material enable precise recognition of DiAcSpm through electrostatic complementarity and molecular conformation matching. These cavities specifically target the acetylated sites and the polyamine backbone of DiAcSpm, while non-target polyamines (e.g., norspermine) fail to form stable complexes due to variations in chain length or charge distribution. Experimental results indicated that DiAcSpm exhibited markedly longer retention times and higher peak area than the other polyamine analogs.

### 3.5. MIM-CME-HPLC-UV method validation

After extracting DiAcSpm using the prepared MIM column, the eluent was quantitatively analyzed using HPLC-UV. To construct the working curve for DiAcSpm, standard solutions at seven different concentration levels were utilized, with each concentration level replicated three times. S3 Fig illustrates the comparison between the NIM and MIM columns in their ability to recognize the analytes. Table 1 demonstrates the excellent linearity (10–500 µM) and correlation coefficients ($R^2 = 0.9984$) achieved through this method. Specifically, the limits of detection (LOD) and quantification (LOQ) for the precise identification of DiAcSpm were determined to be as low as 3.3 µM and 9.8 µM, respectively. All results indicate that the MIM-CME-HPLC-UV method provides high sensitivity and selectivity for detecting DiAcSpm samples. Furthermore, the method exhibited commendable inter- and intra-day precision, with relative standard deviations (RSD) of less than 4.1%

**Table 1. Performance of MIM-CME-HPLC-UV method for DiAcSpm analysis.**

| Analyte | Linear equation[a] | Linear range (µM) | $R^2$ | LOD (µM) | LOQ (µM) | RSD% (n = 5) | |
|---------|-------------------|-------------------|-------|----------|----------|--------------------|--------------------|
| | | | | | | Intra-day precision | Inter-day precision |
| DiAcSpm | y = 0.254x + 12.36 | 10-500 | 0.9984 | 3.3 | 9.8 | 4.1% | 3.8% |

[a]where y is peak area (mV·min) and x is concentration (µM).

(n = 5), underscoring the potential of the hydrophilic amphiphilic MIM column for enriching DiAcSpm. The reusability of the MIM column was assessed over five consecutive extraction-desorption cycles, yielding an extraction recovery RSD of 3.8% and retaining >95% adsorption capacity, which demonstrates excellent operational stability for repeated clinical analyses.

### 3.6. Determination of DiAcSpm in urine samples

The MIM-CME-HPLC-UV method was employed to conduct an enrichment analysis of trace DiAcSpm in urine samples from breast cancer patients and healthy controls, aiming to evaluate the applicability of the developed method. Fig 7A demonstrates that the DiAcSpm peak remained unaffected by other substances, which proved that PC-based materials exhibit a good anti-matrix interference effect in the enrichment of biological samples. Line a indicates that the direct detection of DiAcSpm in the collected urine samples was not feasible. The results from Lines b and d suggest that the MIM method exhibits good efficiency for the enrichment of spermidine, while the comparison of Lines c and d confirms the specificity of the established method for enriching the target substance. This entire process of specific recognition and enrichment is schematically illustrated in Fig 7B. Consequently, as a zwitterionic co-monomer, MPC provides dual functionality: it reduces non-specific adsorption in polymer imprints and enhances surface biocompatibility through hydrophilicity modulation [38]. Furthermore, the preparation processes of all samples were carried out in accordance with the methods in Section 2.6.

The recovery (R) was calculated using the formula: $R(\%) = \frac{C_{measured}}{C_{spiked}} \times 100$, where $C_{measured}$ is the concentration determined by the calibration curve after sample pretreatment, and $C_{spiked}$ is the known fortified concentration. As presented in Table 2, three different concentration levels were selected, yielding satisfactory recoveries ranging from 76.8% to 91.2%. All results, presented as the mean (95% confidence interval, n = 5), were accompanied by RSD values below 7.2%, confirming the method's robustness for practical application in clinical sample analysis. These results suggest that the method holds significant potential and value for practical applications in determining DiAcSpm in both breast cancer patients (C1, C2, C3) and healthy controls (H2, H2, H3).

### 3.7. Method comparison

As detailed in the comparative analysis (S2 Table), the MIM-CME-HPLC-UV method established in this study exhibits distinct advantages for the determination of DiAcSpm. Notably, it eliminates the need for derivatization, thereby simplifying

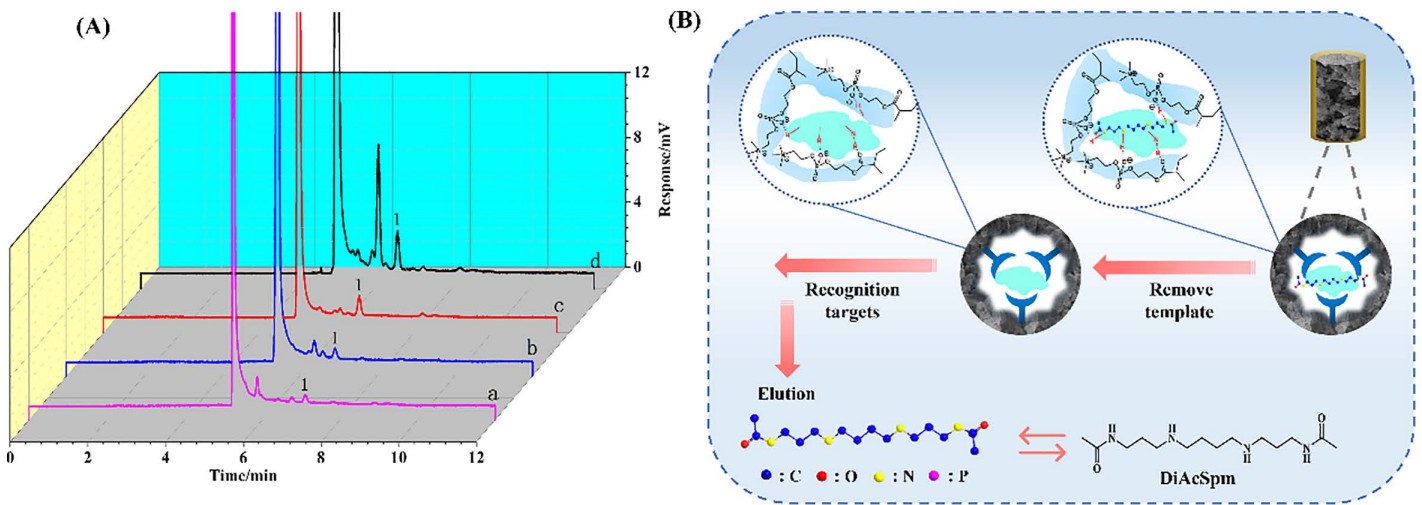

**Fig 7. Detection of DiAcSpm in urine via MIM-CME method and its selective enrichment mechanism.** (A) The HPLC-UV chromatograms of spiked 25 μM DiAcSpm from urine sample. (B) schematic diagram of the process of specific enrichment and elution of MIM-CME polymers and its potential recognition mechanism. Lines: **(a)** direct analysis; **(b)** MIM-CME analysis from healthy control; **(c)** NIM-CME analysis and **(d)** MIM-CME analysis from breast cancer patients. Peak: 1: DiAcSpm.

**Table 2. Method validation data for the determination of DiAcSpm in human urine using the zwitterionic MIM-CME-HPLC-UV method.**

| Sample | Spiked Level (µM) | Recovery, % (95% CI), n = 5 | | RSD, % (n = 5) | |
|---|---|---|---|---|---|
| | | Intra-day | Inter-day | Intra-day | Inter-day |
| C1 | 25 | 76.8 (76.2 - 77.4) | 88.7 (87.9 - 89.5) | 4.8 | 5.7 |
| C2 | 50 | 91.2 (90.8 - 91.6) | 90.6 (90.2 - 91.0) | 2.5 | 3.2 |
| C3 | 100 | 84.5 (83.8 - 85.2) | 86.6 (86.2 - 87.0) | 4.8 | 3.6 |
| H1 | 25 | 82.6 (82.0 - 83.2) | 82.5 (81.8 - 83.2) | 3.2 | 5.2 |
| H2 | 50 | 78.4 (77.4 - 79.4) | 80.5 (80.0 - 81.0) | 7.2 | 3.5 |
| H3 | 100 | 86.3 (85.9 - 86.7) | 87.2 (86.8 - 87.6) | 2.3 | 3.2 |

the sample preparation process relative to several existing approaches. Although the sensitivity (LOD of 3.3 µM) is lower than that of mass spectrometry-based methods, the proposed technique offers an excellent and wide linear range (10–500 µM), superior precision (both intra- and inter-day RSD < 4.1%), and highly consistent recovery rates. A key strength of this approach is the incorporation of a molecularly imprinted polymer (MIM) for selective extraction, which ensures high specificity and reusability (>95% capacity retention after 5 cycles). Combined with the cost-effectiveness and operational simplicity of the HPLC-UV platform, this method represents a robust, reliable, and practical alternative for the routine analysis of specific polyamines such as DiAcSpm in clinical samples.

## 4. Conclusion

The detection of N1,N12-diacetylspermine (DiAcSpm), a critical polyamine a critical polyamine biomarker for breast cancer, poses a significant challenge in aqueous environments due to the inherent weakness of non-covalent interactions in molecularly imprinted polymers (MIPs). This instability hinders the formation of effective imprinting cavities and limits the application of molecularly imprinted monolithic (MIM) column in complex biological environments. To address this limitation, a zwitterionic-based MIM column was prepared using MPC as functional monomers. The hydrophilic group on the MPC facilitates the formation of a strong hydrogen bond complex with DiAcSpm. The superhydrophilic nature of its surface enhances the recognition of target analytes in aqueous solutions, while the biocompatibility afforded by the unique phosphocholine (PC) structure mitigates interference from complex biological matrices. The established MIM-CME-HPLC-UV method exhibited exceptional extraction performance and stability, as confirmed through recovery and precision assessments using real biological samples. Integrating this MIM strategy into sample preparation workflows can valuably complement existing analytical techniques, enabling precise and cost-effective DiAcSpm detection in complex samples and thereby establishing a reliable foundation for the early screening of breast cancer.

## Supporting information

**S1 Fig. EDS image and the element analysis of MIM column.**
(TIF)

**S2 Fig. The molecular structures of spermidine, spermine, and norspermine.**
(PNG)

**S3 Fig. The HPLC-UV chromatograms of (a) blank sample; 25 µM standard DiAcSpm sample on (b) directly injection; (c) extraction with NIM column and (c) extraction with MIM column.** Peak: 1: DiAcSpm.
(DOCX)

**S1 Table. Quantitative evaluation of the MIM selectivity.**
(DOCX)

**S2 Table. Comparative evaluation of qnalytical methods for polyamine quantification.**
(DOCX)

**S1 File. Graphical abstract.**
(TIF)

## Author contributions

**Conceptualization:** Keming Ying.

**Formal analysis:** Keming Ying.

**Funding acquisition:** Keming Ying.

**Investigation:** Keming Ying, Han Xue, Shenheng Du.

**Methodology:** Han Xue, Shenheng Du.

**Software:** Xiaoyun Lei.

**Supervision:** Keming Ying, Xiaoyun Lei.

**Visualization:** Keming Ying.

**Writing – original draft:** Xin Wang.

**Writing – review & editing:** Xiaoyun Lei.

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
