## [Decision Letter · Decision Letter 0]

3 Sep 2025

Dear Dr. Ying,

We look forward to receiving your revised manuscript.

Kind regards,

Shabi Abbas Zaidi, Ph.D.

Academic Editor

PLOS ONE

Journal Requirements:

2. In the ethics statement in the Methods, you have specified that verbal consent was obtained. Please provide additional details regarding how this consent was documented and witnessed, and state whether this was approved by the IRB.

“This work was funded by the Hanzhong Science and Technology Research Project (SKJJKJGG05); Doctoral Studio Program of Hanzhong Central Hospital (YGS24-06).”

“This work was funded by the Hanzhong Science and Technology Research Project (SKJJKJGG05); Doctoral Studio Program of Hanzhong Central Hospital (YGS24-06). “

“This work was funded by the Hanzhong Science and Technology Research Project (SKJJKJGG05); Doctoral Studio Program of Hanzhong Central Hospital (YGS24-06).”

Reviewers' comments:

Reviewer's Responses to Questions

**Comments to the Author**

1. Is the manuscript technically sound, and do the data support the conclusions?

Reviewer #1: Yes

Reviewer #2: Yes

2. Has the statistical analysis been performed appropriately and rigorously?

Reviewer #1: Yes

Reviewer #2: No

3. Have the authors made all data underlying the findings in their manuscript fully available?

Reviewer #1: Yes

Reviewer #2: No

4. Is the manuscript presented in an intelligible fashion and written in standard English?

Reviewer #1: Yes

Reviewer #2: Yes

Reviewer #1: ID: PONE-D-25-32785

Title: Zwitterionic Molecularly Imprinted Polymers for Selective Capillary Microextraction of Spermine from Breast Cancer

This manuscript is suitable for publication in this journal after major revisions.

1. How stable is the MIM column over multiple extraction cycles? Can you provide a column reusability study?

2. Add statistical analysis (e.g., t-test or ANOVA) to compare MIM and NIM performance with significance levels.

3. Clearly define the recovery calculation formula and add confidence intervals to recovery and precision data.

4. Discuss clinical relevance of the LOD/LOQ values in the context of known spermine/DiAcSpm concentrations in patient populations.

5. Incorporate a visual workflow figure (graphical abstract) summarizing the CME process and advantages.

6. Expand on the novelty of your work—how is your approach substantially different from previous MIP or zwitterionic-MIP studies?

7. Compare your results (LOD, linear range, recovery) with other published methods in a concise benchmarking table.

8. Clarify whether the optimized extraction parameters were validated across multiple MIM batches.

9. How does the binding affinity of the MIM column vary with pH or ionic strength?

10. Improve figure readability (especially SEM and BET plots) by labeling axes clearly and adding scale bars and annotations.

11. Were matrix effects quantitatively evaluated (e.g., via matrix-matched calibration or recovery experiments)?

Reviewer #2: In this manuscript, the capillary-confined molecularly imprinted monolithic (MIM) column was proposed and used for the selective recognition of spermine in the urine of breast cancer patients.

This work is interesting for breast cancer diagnosis. However, there are still some issues to be addressed. The following changes are suggested

1、1�Figure 3.( A)、(B) lacks a title for the x-axis.�2�In figure 5.( A)、(B), the y-axis values lacks numerical labels, and the x-axis “Relative pressive�P/P0�” is recommended to revise to “Relative pressure�P/P0�”�3�The y-axis in figure 7.(A) is missing its unit label.

2、The meanings of the parameters in the equation given on line 178 are not provided, and the manuscript does not demonstrate application of this equation.

3、The interpretation of the characterization result is somewhat rough. It is recommended to revise this section and explicitly list key parameters, such as total surface area, pore diameter, and total pore volume—to clearly demonstrate the superiority of the MIM.

4、Lines 232-233 of the manuscript describe that “both columns exhibit a continuous porous structure with an average diameter of approximately 5.0 μm”, and subsequent the pore size distribution curves were interpreted that “the pore size of NIM is noticeably smaller than that of MIM”. Please explain this paradoxical statement.

5、Basing conclusions solely on the material’s specific surface area and total pore volume may be one-sided; it is advisable to conduct additional study to evaluate its adsorption performance more thoroughly.

6、Parameter usage should be consistent throughout the manuscript; for adsorption capacity, “Qc” and “Q occurred in different places.

7、Lines 342–346: the y-axis in Figure 7.( A)、(B) is labeled “peak area,” which is inconsistent with the text's description of “peak response values”. In addition, can the unit for “peak area” be expressed as mAU? Please check this issue in all relevant figures.

8、I have some confusion regarding the determination of DiAcSpm in the samples described in Section 3.6 of the manuscript, and uncertain what authors' target analyte actually is.

9、The citation format of the references is inconsistent (e.g., the volume number in reference 3 is incorrect, and the page numbers in reference 5 are wrong); please check all references carefully.

10. The target analyte in the article is spermine, but the template used in the preparation of MIM is N1,N12-diethylsperminetetrahydrochloride. However, there is no clear explanation in the article why the analyte is spermine but N1,N12-diethylsperminetetrahydrochloride is selected as the template?

11. In the preparation of molecularly imprinted monolithic column, the dosage of functional monomer and template molecule is weight ratio, while the other parts of this paper are mainly measured by mole ratio. In addition, Methanol and tetrahydrofuran are liquids under normal conditions, please take volume as the unit.

12. How long is the prepolymerization time of molecularly imprinted monolithic column? It is suggested to supplement.

13. Why does the polymerization solution in the preparation of molecularly imprinted monolithic columns not need to be deoxygenated? Does oxygen have no effect on the properties of materials?

14. As an important parameter, the evaluation equation of MIM enrichment ability in 2.5 is suggested to be written in detail in this paper.

15. Regarding the selection of washing solvent in 3.1, MeOH:HCl or MeOH:NaOH as eluting solvent has only one ratio. It is recommended to increase the study of different elution ratios to increase the persuasiveness of the study.

16. The molecular structure and detection concentration of structural analogues of N1,N12-diethylsperminetetrahydrochloride in 3.4.2 are suggested to be given in the article; evaluation parameters are also given to quantify the selectivity advantage.

17. Information about linear equations should be added to Table to improve methodological parameters.

18. There are few pairs between this detection method and existing breast cancer screening methods. It is suggested to increase the comparison between this method and existing clinical methods to highlight the superiority and innovation of this method.

19. It is suggested to supplement the research data on the service life of MIM column to improve the practicability of this method.

20. In fact, the sensitivity is not well in the proposed method. Is the method suitable for medical diagnosis? Please provide the background knowledge and data for the target analyte detection, especially in clinic analysis.

**Do you want your identity to be public for this peer review?** For information about this choice, including consent withdrawal, please see our Privacy Policy

Reviewer #1: No

Reviewer #2: No

---

## [Author Response · Author response to Decision Letter 1]

30 Nov 2025

1. Style Requirements

Response: We have carefully revised the manuscript format, including the title page, main text, reference style, and figure/file naming, to ensure full compliance with PLOS ONE's style requirements.

2. Ethics Statement

Response: We have revised the ethics statement in the Methods section (Section 2.6) to provide more details. The statement now reads:

"Verbal informed consent was obtained from all participants prior to sample collection. This consent procedure, including the process of documentation and witnessing, was specifically reviewed and approved by the Institutional Review Board (IRB) of Hanzhong Central Hospital (Approval No. [2024] Ethics Review No. 32 ). All methods were performed in accordance with the relevant guidelines and regulations."

3. Financial Disclosure

Response: We confirm that the funders had no role in our study. The following statement has been prepared for inclusion in the cover letter and will be updated in the online submission form as requested: "The funders had no role in study design, data collection and analysis, decision to publish, or preparation of the manuscript."

4. Data Availability Statement

Response: We confirm that all data required to replicate the findings of this study are now included within the manuscript and its Supporting Information files. This includes the raw data points behind all means, standard deviations, graphs, and analyses presented in the study.

5. ORCID iD

Response: We confirm that the corresponding author has a valid ORCID iD and it has been validated in the Editorial Manager system.

6. Funding Statement in Acknowledgments

Response: The funding information has been removed from the Acknowledgments section of the manuscript. The Funding Statement for the online submission form remains as:

"This work was funded by the Hanzhong Science and Technology Research Project (SKJJKJGG05); Doctoral Studio Program of Hanzhong Central Hospital (YGS24-06)."

Reviewer #1: ID: PONE-D-25-32785

Title: Zwitterionic Molecularly Imprinted Polymers for Selective Capillary Microextraction of Spermine from Breast Cancer

This manuscript is suitable for publication in this journal after major revisions.

Comment 1. How stable is the MIM column over multiple extraction cycles? Can you provide a column reusability study?

Response: Thank you for pointing this out. We evaluated the reusability of the MIM column by performing 5 consecutive extraction-desorption cycles using a 200 μM DiAcSpm standard solution. The column retained >95% of its initial adsorption capacity after 5 cycles, with a relative standard deviation (RSD) of 3.8% for the extraction recovery. These results demonstrate excellent operational stability and reusability of the MIM column, making it suitable for repeated analyses in clinical sample processing. The data have been added to Section 3.5 of the revised manuscript. Please find the results in page 18, lines 395-398�The reusability of the MIM column was assessed over five consecutive extraction-desorption cycles, yielding an extraction recovery RSD of 3.8% and retaining >95% adsorption capacity, which demonstrates excellent operational stability for repeated clinical analyses.

Comment 2. Add statistical analysis (e.g., t-test or ANOVA) to compare MIM and NIM performance with significance levels.

Response: Thank you very much for your constructive suggestions. We have performed an unpaired t-test to statistically compare the adsorption capacities of MIM and NIM columns (n=5). The MIM column showed a significantly higher adsorption capacity (67.45 ± 2.84 μmol/g) compared to the NIM column (38.54 ± 1.92 μmol/g), with a p-value < 0.001. This indicates a highly significant difference in performance, confirming the effectiveness of the molecular imprinting process. The statistical results are now included in Section 3.4.1. on page 17, lines 357-359: This difference was statistically confirmed by an unpaired t-test (p< 0.001, n = 5), indicating the successful creation of specific binding sites in the MIM.

Comment 3. Clearly define the recovery calculation formula and add confidence intervals to recovery and precision data.

Response: Thank you very much for your constructive suggestions. The recovery (R) was calculated using the formula: R(%)=C_measured/C_spiked ×100 , where Cmeasured is the concentration determined by the calibration curve after sample pretreatment, and Cspiked is the known fortified concentration. Confidence intervals (95% CI) have been added to the recovery and precision data in Table 2. Please check the modification in the latest manuscript on pages 20-21, lines 422-431: The recovery (R) was calculated using the formula: R(%)=C_measured/C_spiked ×100, where Cmeasured is the concentration determined by the calibration curve after sample pretreatment, and Cspiked is the known fortified concentration. As presented in Table 2, three different concentration levels were selected, yielding satisfactory recoveries ranging from 76.8% to 91.2%. All results, presented as the mean (95% confidence interval, n=5), were accompanied by RSD values below 7.2%, confirming the method's robustness for practical application in clinical sample analysis. These results suggest that the method holds significant potential and value for practical applications in determining DiAcSpm in both breast cancer patients (C1, C2, C3) and healthy controls (H2, H2, H3).

Table 2. Method validation data for the determination of DiAcSpm in human urine using the zwitterionic MIM-CME-HPLC-UV method.

Sample Spiked Level (μM) Recovery, % (95% CI), n=5 RSD, % (n=5)

Intra-day Inter-day Intra-day Inter-day

C1 25 76.8 (76.2 - 77.4) 88.7 (87.9 - 89.5) 4.8 5.7

C2 50 91.2 (90.8 - 91.6) 90.6 (90.2 - 91.0) 2.5 3.2

C3 100 84.5 (83.8 - 85.2) 86.6 (86.2 - 87.0) 4.8 3.6

H1 25 82.6 (82.0 - 83.2) 82.5 (81.8 - 83.2) 3.2 5.2

H2 50 78.4 (77.4 - 79.4) 80.5 (80.0 - 81.0) 7.2 3.5

H3 100 86.3 (85.9 - 86.7) 87.2 (86.8 - 87.6) 2.3 3.2

Comment 4. Discuss clinical relevance of the LOD/LOQ values in the context of known spermine/DiAcSpm concentrations in patient populations.

Response: Thank you very much for your constructive suggestions. Reported demonstrate that urinary levels of N1,N12-diacetylspermine (DiAcSpm) and N1,N12-diacetylspermidine (DiAcSpd) were significantly elevated in breast cancer patients compared to healthy controls, with a substantial proportion of patients exhibiting values well above the established cut-off lines, particularly in those with advanced-stage disease. In terms of diagnostic performance, DiAcSpm emerged as the most promising marker, demonstrating a sensitivity of 46.4% for breast cancer detection, which was markedly superior to the sensitivity of conventional serum tumor markers carcinoembryonic antigen and CA 15-3. The study-defined cut-off values, reported in nmol per gram creatinine, were 298 nmol/g cre for DiAcSpm and 857 nmol/g cre for DiAcSpd. When estimated based on an average daily urine output of 1-2 liters in adults, these cut-off values correspond approximately to volumetric molar concentrations of 150-300 nM for DiAcSpm and 430-850 nM for DiAcSpd, collectively indicating that urinary DiAcSpm concentration represents a highly promising non-invasive tumor marker for breast cancer. This confirms that our method is sufficiently sensitive for clinical detection and monitoring. The above content has been added to the manuscript. Please check the modification in the latest manuscript on page 3, lines 58-63: Urinary N1,N12-diacetylspermine (DiAcSpm) is significantly elevated in breast cancer patients and possesses considerable potential as a non-invasive diagnostic biomarker, with an estimated cut-off of 150-850 nM and a sensitivity of 46.4% that surpasses conventional serum biomarkers, as measured by competitive ELISA (Enzyme-Linked Immunosorbent Assay) [12].

Reference

[12] Umemori Y, Ohe Y, Kuribayashi K, Tsuji N, Nishidate T, Kameshima H, Hirata K, Watanabe N (2010) Evaluating the utility of N1,N12-diacetylspermine and N1,N8-diacetylspermidine in urine as tumor markers for breast and colorectal cancers. Clinica Chimica Acta 411:1894-1899

Comment 5. Incorporate a visual workflow figure (graphical abstract) summarizing the CME process and advantages.

Response: Thank you very much for your constructive suggestions. A graphical abstract has been added to the manuscript, illustrating the key steps of MIM preparation, CME extraction, and HPLC-UV analysis. The figure highlights the advantages of the zwitterionic MIM, including high selectivity, anti-fouling properties, and compatibility with complex biological samples.

Comment 6. Expand on the novelty of your work—how is your approach substantially different from previous MIP or zwitterionic-MIP studies?

Response: Thank you very much for your constructive suggestions. This study presents the first integration of a zwitterionic MPC monomer into a capillary molecularly imprinted monolith (MIM) for highly selective extraction of spermine/diacetylspermine (DiAcSpm). Although molecularly imprinted materials have been explored for biological sample detection, the application of zwitterionic MIMs in complex bodily fluids remains relatively underdeveloped. Unlike conventional molecularly imprinted polymers (MIPs), which often exhibit non-specific binding in aqueous matrices, the developed zwitterionic MIM not only provides specific molecular recognition but also demonstrates excellent anti-fouling properties. This enables direct and highly selective extraction from urine without the need for derivatization. By effectively addressing key limitations in existing polyamine analysis—such as insufficient selectivity, matrix interference, and operational complexity—this approach offers a more practical and robust pathway for efficient detection of polyamines in bodily fluids. The above description has been explained in the Introduction on pages 4-5, lines 77-99.

Comment 7. Compare your results (LOD, linear range, recovery) with other published methods in a concise benchmarking table.

Response: Thank you very much for your constructive suggestions. We have added a section 3.7 on method comparison and the details added in Table S2 in the Supporting Information. Please check the modification in the latest manuscript on page 21, lines 434-447.

3.7 Method comparison

As detailed in the comparative analysis (Table S2), the MIM-CME-HPLC-UV method established in this study exhibits distinct advantages for the determination of DiAcSpm. Notably, it eliminates the need for derivatization, thereby simplifying the sample preparation process relative to several existing approaches. Although the sensitivity (LOD of 3.3 μM) is lower than that of mass spectrometry-based methods, the proposed technique offers an excellent and wide linear range (10–500 μM), superior precision (both intra- and inter-day RSD < 4.1%), and highly consistent recovery rates. A key strength of this approach is the incorporation of a molecularly imprinted polymer (MIM) for selective extraction, which ensures high specificity and reusability (>95% capacity retention after 5 cycles). Combined with the cost-effectiveness and operational simplicity of the HPLC-UV platform, this method represents a robust, reliable, and practical alternative for the routine analysis of specific polyamines such as DiAcSpm in clinical samples.

Comment 8. Clarify whether the optimized extraction parameters were validated across multiple MIM batches.

Response: Thank you for raising this important point. We confirm that the extraction parameters were indeed optimized and validated using three independently prepared batches of the SPME material to ensure the method's reproducibility and reliability. As you have rightly noted, the error bars presented in Figure 5 reflect the performance variations observed across these three batches. To clarify this more explicitly, we have now added the label “n = 3” to Figure 5 in the revised manuscript and supplemented the corresponding text in the Methods section with the statement: “The experiments were conducted in triplicate (n=3).”

Fig 5. Effects of MIM-CME conditions on the extraction performance of DiAcSpm. (A) ACN content in loading solution (%); (B) ACN content in eluent (%); (C) formic acid content in eluent (%); (D) flow rate of loading solvent (μL/min); (E) volume of sample loading (mL). All experiments were performed in triplicate (n=3).

Comment 9. How does the binding affinity of the MIM column vary with pH or ionic strength?

Response: Thank you for this insightful question. We have systematically investigated the influence of formic acid content (i.e., pH) in the eluent on the desorption behavior of DiAcSpm to elucidate the regulatory mechanism of binding affinity. As shown in Figure 6C, the peak area of DiAcSpm increased significantly as the formic acid content rose from 0.1% to 0.2%, reaching a maximum at 0.2%. This indicates that under this mildly acidic condition, the analyte is sufficiently protonated and an optimal balance in ionic interaction with the stationary phase is achieved, thereby facilitating efficient desorption. When the formic acid content was further increased to 0.3%–0.5%, the peak area gradually decreased, which we attribute to the excessively acidic environment potentially causing over-protonation or altering the solvation equilibrium, thereby hindering complete desorption.

Regarding the effect of ionic strength, it should be noted that no additional inorganic salts were introduced in this phase of experiments aimed at optimizing the eluent composition. Therefore, our current results primarily reflect the regulation of binding affinity by pH (formic acid content), while the direct impact of ionic strength was not investigated. This design also suggests the potential of our method to exhibit certain resistance to salt interference in real samples (e.g., urine).

In summary, we confirm that the pH of the eluent is a key parameter regulating the binding affinity of the MIM for DiAcSpm, and the optimal extraction performance was achieved under the condition of 0.2% formic acid. Please check the modification in the latest manuscript on pages 14-15, lines 305-319.

3.3.3 Optimization of pH on eluent

Furthermore, the pH of the eluent, governed by the formic acid content, significantly influences the desorption process. Although the zwitterionic monomer MPC maintains a net negative charge over a wide pH range, facilitating cation-exchange interactions, the strength of these interactions is modulated by the ionic environment. As shown in Fig 5C, the peak area of DiAcSpm initially increased as the formic acid content rose from 0.1% to 0.2%, reaching a maximum. This could be explained by the sufficient protonation of DiAcSpm, which strengthens its ionic interaction with the stationary phase during the loading step and allows for efficient release in a mild acidic eluent. However, beyond 0.2%, a higher formic acid content (0.3%–0.5%) resulted in a gradual decrease in the peak area. This decline is likely due to an excessively acidic environment causing overly strong protonation or altering the solvation equilibrium, thereby hampering complete desorption. Based on these observations, an eluent consisting of 0.2% formic acid in eluent solution was identified as the optimal condition.

Comment 10. Improve figure readability (especially SEM and BET plots) by labeling axes clearly and adding scale bars and annotations.

Response: Thank you very much for your constructive suggestions. All figures (SEM, BET, etc.) have been revised to include clear axis labels, scale bars, and descriptive annotations. Fig. 3 now features scale bars for SEM images, and Fig. 4 includes well-labeled axes and units for BET plots.

Fig 3. SEM micrographs of (A and B) NIM and (C and D) MIM column before template removal. Magnification: A and C: 1,000×; B and D: 5,000×

Fig 4. Specific surface area and pore size distribution curve (insert) of the (A) NIM and (B) MIM column.

Comment 11. Were matrix effects quantitatively evaluated (e.g., via matrix-matched calibration or recovery experiments)?

Response: Thank you for this insightful question. The potential matrix effect was evaluated indirectly via recovery experiments using spiked samples. The

---

## [Decision Letter · Decision Letter 1]

11 Dec 2025

Zwitterionic Molecularly Imprinted Polymers for Selective Capillary Microextraction of N1,N12-Diacetylspermine (DiAcSpm) from Breast Cancer

PONE-D-25-32785R1

Dear Dr. Ying,

We’re pleased to inform you that your manuscript has been judged scientifically suitable for publication and will be formally accepted for publication once it meets all outstanding technical requirements.

Within one week, you’ll receive an email detailing the required amendments. When these have been addressed, you’ll receive a formal acceptance letter, and your manuscript will be scheduled for publication.

Kind regards,

Shabi Abbas Zaidi, Ph.D.

Academic Editor

PLOS One

Additional Editor Comments (optional):

No comments

Reviewers' comments:

Reviewer's Responses to Questions

**Comments to the Author**

Reviewer #1: All comments have been addressed

Reviewer #2: All comments have been addressed

2. Is the manuscript technically sound, and do the data support the conclusions?

Reviewer #1: Yes

Reviewer #2: Yes

3. Has the statistical analysis been performed appropriately and rigorously?

Reviewer #1: Yes

Reviewer #2: N/A

4. Have the authors made all data underlying the findings in their manuscript fully available?

Reviewer #1: Yes

Reviewer #2: Yes

5. Is the manuscript presented in an intelligible fashion and written in standard English?

Reviewer #1: Yes

Reviewer #2: Yes

Reviewer #1: The author has responded well to all the comments, the changes are well indicated, and the manuscript is acceptable as is.

Reviewer #2: The authors have done a good revision based on the suggestions. I think it can be accepted now. Thanks.

**Do you want your identity to be public for this peer review?** For information about this choice, including consent withdrawal, please see our Privacy Policy

Reviewer #1: No

Reviewer #2: No

---

## [Editor Report · Acceptance letter]

PONE-D-25-32785R1

PLOS One

Dear Dr. Ying,

I'm pleased to inform you that your manuscript has been deemed suitable for publication in PLOS One. Congratulations! Your manuscript is now being handed over to our production team.

Kind regards,

on behalf of

Dr. Shabi Abbas Zaidi

Academic Editor

PLOS One